# The potential of tumour microenvironment markers to stratify the risk of recurrence in prostate cancer patients

Thomas Gevaert[1,2,3,4]☯*, Yves-Rémi Van Eycke[5,6]☯, Thomas Vanden Broeck[1,4,7], Hein Van Poppel[1,2,4], Isabelle Salmon[6,8,9], Sandrine Rorive[8,9], Tim Muilwijk[1,2], Frank Claessens[4,7], Dirk De Ridder[1,2,4], Steven Joniau[1,2,4‡], Christine Decaestecker[5,6‡]

1 Department of Urology, UZ Leuven, Leuven, Belgium, 2 Organ Systems, KU Leuven, Leuven, Belgium, 3 Department of Pathology, AZ Klina, Brasschaat, Belgium, 4 P.E.A.R.L. (ProstatE cAncer Research Leuven), Leuven, Belgium, 5 Laboratories of Image, Synthesis and Analysis (LISA), Brussels School of Engineering/École polytechnique de Bruxelles, ULB, Brussels, Belgium, 6 DIAPath—Center for Microscopy and Molecular Imaging (CMMI), ULB, Gosselies, Belgium, 7 Department of Molecular and Cellular Medicine, KU Leuven, Leuven, Belgium, 8 Department of Pathology, Erasme University Hospital, Université Libre de Bruxelles (ULB), Brussels, Belgium, 9 Centre Universitaire Inter Régional d'Expertise en Anatomie Pathologique Hospitalière (CurePath), Jumet, Belgium

☯ These authors contributed equally to this work.
‡ These authors also contributed equally to this work.
* Gevaert.thomas1979@gmail.com

**Data Availability Statement:** All relevant data are within the paper and its Supporting Information files.

## Abstract

The tumour micro-environment (TME) plays a crucial role in the onset and progression of prostate cancer (PCa). Here we studied the potential of a selected panel of TME-markers to predict clinical recurrence (CLR) in PCa. Patient cohorts were matched for the presence or absence of CLR 5 years post-prostatectomy. Tissue micro-arrays (TMA) were composed with both prostate non-tumour (PNT) and PCa tissue and subsequently processed for immunohistochemistry (IHC). The IHC panel included markers for cancer activated fibroblasts (CAFs), blood vessels and steroid hormone receptors ((SHR): androgen receptor (AR), progesterone receptor (PR) and estrogen receptor (ER)). Stained slides were digitalised, selectively annotated and analysed for percentage of marker expression with standardized and validated image analysis algorithms. A univariable analysis identified several TME markers with significant impact on CR: expression of CD31 (vascular marker) in PNT stroma, expression of alpha smooth muscle actin (αSMA) in PCa stroma, and PR expression ratio between PCa stroma and PNT stroma. A multivariable model, which included CD31 expression (vascular marker) in PNT stroma and PR expression ratio between PCa stroma and PNT stroma, could significantly stratify patients for CLR, with the identification of a low risk and high-risk subgroup. If validated and confirmed in an independent prospective series, this subgroup might have clinical potential for PCa patient stratification.

**Funding:** This work was supported by the Josef De Wever Fonds (Leuven, Belgium) and the Foundation Yvonne Boël (Brussels, Belgium). The CMMI is supported by the European Regional Development Fund and the Walloon Region (Belgium). FC and SJ received grant GOA/15/017 from the KU Leuven Research Council. CD is a senior research associate supported by the Fonds National de la Recherche Scientifique (Brussels, Belgium).

**Competing interests:** The authors have declared that no competing interests exist.

## Introduction

Stratification of cancer patients based on molecular expression profiles will become crucial in the era of personalised medicine and reliable biomarkers will be pivotal to ensure optimal patient selection. Three major groups of biomarkers can be distinguished: diagnostic biomarkers refer to a biological parameter that aids the diagnosis of a disease; prognostic biomarkers inform about a likely cancer outcome (e.g., disease recurrence, disease progression, death) independent of treatment received, while a biomarker is predictive or theragnostic if the treatment effect (experimental compared with control) is different for biomarker-positive patients compared with biomarker-negative patients [1].

During the last decade, many predictive and prognostic biomarkers have entered clinical oncological practice, and many more are expected to do so in the years to come. The use of biomarkers to stratify patients with prostate cancer (PCa) is limited, although several clinical trials are exploring potential candidates [2]. Nevertheless, some commercial prognostic biomarker-panels to stratify patients with PCa have already entered clinical practice [2].

Tumour cells have always been the natural candidates to explore new biomarkers. However, the recent evolutions in immunomodulatory and anti-angiogenic drugs have also revealed a pivotal role for the tumour micro-environment (TME) in the path toward personalised cancer care [3–5]. This TME consists of a complex network of immune cells, stromal fibroblasts, blood vessels, pericytes, mesenchymal stem cells (MSCs), neural cells, fat cells, and secreted soluble and insoluble factors [6, 7]. The interplay between the TME and tumour cells is complex and contributes to the gradual transformation of normal cells towards neoplastic cancer cells [7].

There were a myriad of PCa biomarker studies focusing on the TME over the last 10–15 years, and many of them have focused on its potential prognostic value. Hence, clinical recurrence (CLR) in PCa was linked to changes in the TME in several studies [8–12], including reports on CLR-dependent alterations in expression profiles of steroid hormone receptors (SHR) [13–16], cancer activated fibroblast (CAF) markers [17, 18] and vascular markers. Interestingly, most commercially available prognostic biomarker-panels have also several stromal cell markers in their gene panels [2]. However, transversal data on the relation between CLR and different key markers of the TME are often lacking, since most of these studies focus on individual markers and/or single pathways.

Previously, we investigated the relation between different GS and the expression of several TME cell markers [19]. This panel included the vascular marker CD31, CAF markers (CD34, caveolin-1 (CAV-1) and alpha smooth muscle actin (αSMA)) and steroid hormone receptors (SHR: androgen receptor (AR), progesterone receptor (PR) and estrogen receptor alpha (ERα)) in paired prostate non-tumour (PNT) and PCa tissue. In this study we investigated the same panel of TME-markers on its prognostic potential to predict CLR in PCa. However, we restricted our analyses to matched cohorts of patients (with or without CLR), all with GS7 tumours, in order to focus on highlighting the prognostic contributions of the markers analysed.

## Methods

### Patient samples

Institutional review board approval for a retrospective analysis of archival biobank tissue was obtained from the institution's ethics committee (Ethische commissie onderzoek UZ / KU Leuven), together with ethical agreement (approval number S55860). All patient-related sample-data were fully anonymized in the study analysis. Patient's medical records and tissue

blocks were assessed during the period June 2016-June 2018. Since it was a retrospective study on archival tissues, the institution's ethics committee waived the requirement for informed consent. The source of the medical records/samples analyzed in this work was the University Hospitals Leuven (UZ Leuven).

Matched cohorts (n = 24 per group) were composed of patients with and without CLR after radical prostatectomy. CLR was defined as local and/or distant disease recurrence—as established with PSA-relapse combined with confirmed lesions on CT scan, bone scan or PSMA-PET—and minimal follow-up time was 5 years. From the 24 patients with CLR, 14 showed distant metastasis, 9 showed local recurrence and 1 presented with both. Patient cohorts were partially overlapping with cohorts described previously [19] and were matched for Grade groups (GG), lymph node-status, margin-status, p-stage and age (see Table 1 for clinicopathological parameters). All cases were revised for correct ISUP GG by an experienced urogenital pathologist (TG) and only GS7 (ISUP GG2 / GG3) cases without tertiary component were included. Paired PNT samples with histologically normal prostate tissue were collected from tissue blocks from 36 patients of the same cohorts of which 16 presented with CLR.

## Sample handling

Some parts of the methods described below are identical to those used in a previous study [19]. Identical parts apply to the design of the tissue micro-arrays (TMA), immunohistochemistry and data analysis methodology.

**Table 1. Clinical and pathological characteristics of patient cohorts.**

| | Total | | Clinical recurrence | | No clinical recurrence | | Prognosis |
|---|---|---|---|---|---|---|---|
| | | | | | | | p-value |
| | Med | IQR | Med | IQR | Med | IQR | |
| Follow-up* (months) | 105 | 62–152 | 81 | 46–125 | 132 | 98–154 | |
| Age (years) | 65 | 59–70 | 63.5 | 57.5–67 | 68 | 60.5–72 | 0.13 |
| | No. | % | No. | % | No. | % | |
| Tumor stage | | | | | | | 0.28 |
| T2 | 10 | 21 | 4 | 17 | 6 | 25 | |
| T3a | 24 | 50 | 10 | 42 | 14 | 58 | |
| T3b | 12 | 25 | 8 | 33 | 4 | 17 | |
| T4 | 2 | 4 | 2 | 8 | 0 | 0 | |
| Nodal stage | | | | | | | NA |
| Negative (N0) | 46 | 96 | 22 | 92 | 24 | 100 | |
| Positive (N+) | 2 | 4 | 2 | 8 | 0 | 0 | |
| ISUP Grade Group | | | | | | | 0.61 |
| GG2 | 20 | 42 | 11 | 46 | 9 | 38 | |
| GG3 | 28 | 58 | 13 | 54 | 15 | 62 | |
| Surgical margins | | | | | | | 0.54 |
| Positive | 20 | 42 | 11 | 46 | 9 | 38 | |
| Negative | 28 | 58 | 13 | 54 | 15 | 62 | |
| Total | 48 | 100 | 24 | 50 | 24 | 50 | |

Med: Median; IQR: inter-quartile range. The p-values for evaluating the prognostic impact were computed by univariate cox regression for age, Chi2 test (multiple group) for stage after grouping T3b and T4, and log-rank test for the other binary features. Because of the matched cohorts, p-values > 0.05 were expected.

*Follow-up indicates the time to recurrence or the complete follow-up in the absence of recurrence

**Tissue micro-arrays.** Layout designs were used to develop the tissue micro-arrays (TMAs) from donor paraffin blocks. Tissue paraffin blocks were retrieved from our institutional database with archived radical prostatectomy specimens. Haematoxylin and eosin (H&E) stained slides were used to select two representative paraffin blocks per patient: one with histologically normal prostate (i.e. PNT) tissue and one with PCa. Cylindrical tissue cores (size: 6mm) were harvested from the paraffin blocks and placed into a recipient paraffin block with the Alphelys minicore technology (Alphelys, France). Three TMAs were required to include all cores from the 48 PCa patients, containing paired PCa and PNT samples. Per patient 6 tumor cores and 6 normal tissue cores were included. Since almost all tumors consisted of peripheral zone PCa, control tissue was taken from the contralateral PNT peripheral zone.

**Immunohistochemistry (IHC).** Prior to enrolment in the study, antibody-clones were carefully selected for their epitope-selectivity and validated on control tissue for staining specificity and reliability (Table 2). Most of the antibodies used in this study are well approved for clinical diagnostic practice (www.nordiqc.com). Antibodies were directed against the vascular marker CD31, CAF markers (CD34, Cav-1 and αSMA) and SHR (AR, PR and ER) (Fig 1 for representative stains).

TMA paraffin blocks were cut in serial slides (5μm) to secure similar tissue areas for all TME IHC markers. IHC stains were performed on the Leica Bond-Max system (Leica Microsystems, Belgium). The automated staining procedure was identical as described previously [19].

**Compartmentalized and quantitative staining analysis.** We followed exactly the same methodology than the one described in our previous study [19]. Briefly, within 2 weeks after staining the TMA slides were digitized at 20x using a calibrated whole slide scanner (Nano-Zoomer 2.0-HT, Hamamatsu, Hamamatsu City, Japan). Using the Visiopharm software package (Visiopharm, Hoersholm, Denmark), an experienced urogenital pathologist (TG) annotated the serial virtual slides to distinguish between epithelial and stromal expression in PCa and PNT samples, while ensuring that similar areas were selected from one marker to another and excluding areas with non-specific staining (e.g. intraluminal) and inflammatory cells. In addition, for CD34, αSMA and Cav-1, blood vessel components were excluded from the stromal areas submitted to analysis.

**Table 2. Properties of the antibody clones used.**

| Immunogen | Clone | Manufacturer/Code | Host | Titer | Control |
|---|---|---|---|---|---|
| *Alpha-smooth muscle actin (α-sma)* | 1A4 | Agilent Technologies, Diegem, Belgium | Mouse | Ready to use | Appendix |
| (N-terminal synthetic decapeptide of α-smooth muscle actin) | | *IR611* | | | |
| *Androgen receptor (AR)* | AR441 | Agilent Technologies, Diegem, Belgium | Mouse | 1/100 | Prostate, Breast (non-tumour) |
| (synthetic peptide with amino acids 229–315 of the human AR) | | *M3562* | | | |
| *Caveolin-1 (Cav-1)* | N20 | Santa-Cruz Biotechnology, Heidelberg, Germany | Rabbit | 1/100 | Lung |
| (synthetic peptide at the N-terminus of human caveolin-1) | | *SC-894* | | | |
| *CD31* | JC70A | Agilent Technologies, Diegem, Belgium | Mouse | Ready to use | Appendix |
| (cell membrane from spleen) | | *IR610* | | | |
| *CD34* | QBend10 | Agilent Technologies, Diegem, Belgium | Mouse | Ready to use | Appendix |
| (endothelial cell membranes from human placenta) | | *IR632* | | | |
| *Estrogen receptor alpha (ER)* | 1D5 | Agilent Technologies, Diegem, Belgium | Mouse | Ready to use | Uterine cervix |
| Soluble recombinant human estrogen receptor | | *IS657* | | | |
| *Progesterone receptor (PR)* | PGR636 | Agilent Technologies, Diegem, Belgium | Mouse | Ready to use | Uterine cervix |
| Full length A-form of human progesterone receptor | | M3569 | | | |

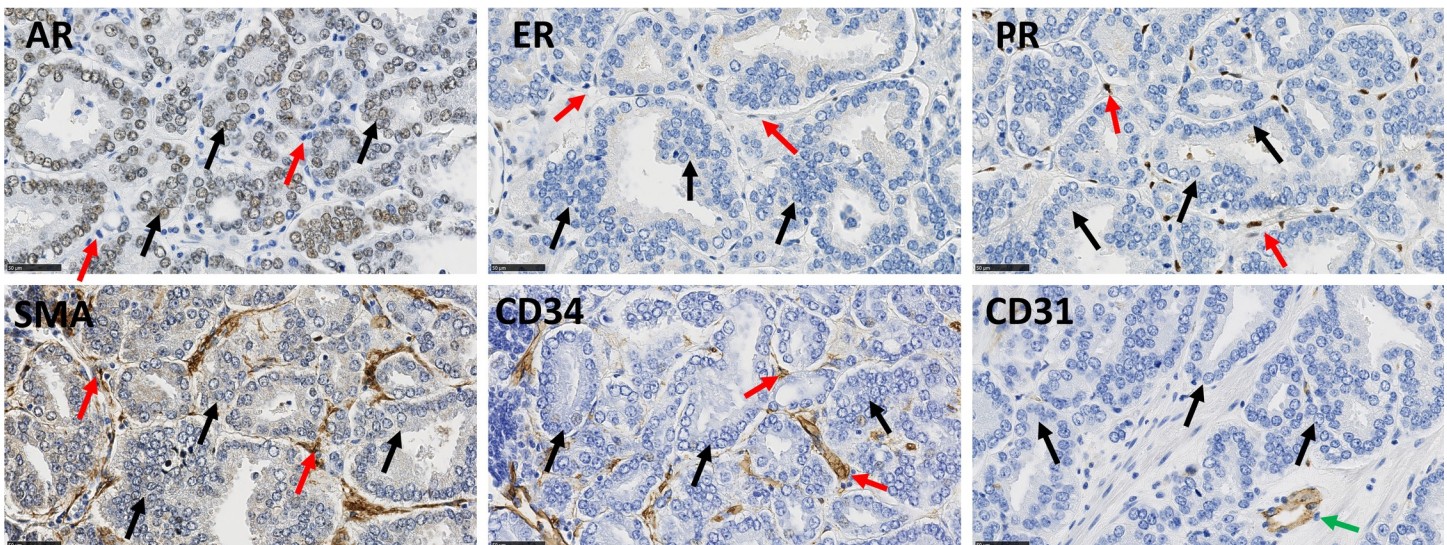

**Fig 1. Illustrative immunohistochemical stains for AR, PR, ER, αSMA, CD34 and CD31 in PCa samples from a single patient.** Black arrows indicate epithelial cells and red arrows indicate stromal cells. AR and ER can be expressed in epithelial and stromal cells; PR is only expressed in stromal cells. αSMA and CD34 are stromal cell markers. CD31 is a vascular marker (green arrow). Scale bar equals 50μm.

For each IHC marker we computed per patient (i.e. by pooling the available TMA cores) the labelling index (LI) within the annotated areas in PCa and PNT samples, using the Visiomorph software package (Visiopharm, Hoersholm, Denmark). For the IHC markers with cytoplasmic expression (CD31, CD34, αSMA, Cav-1), the LI is the percentage of the positive (i.e. immunostained) tissue area, whereas for the IHC markers with nuclear expression (AR, PR, ER) the LI is computed as the percentage of positive pixels in the nuclear area only [19, 20]. High LI values are indicative of high percentages of positive cells. In addition, we computed ratios of LI values between PCa and PNT samples from the same patient, to take into account the proportional variation in PCa with respect to the basal expression level in PNT, and also between epithelial and stromal tissue components when available.

## Statistics

Clinical recurrence-free survival (CRFS) was calculated as the time between surgical intervention and the detection of clinical recurrence. Univariable survival analyses were carried out using the log-rank test and the univariable Cox regression method for qualitative and quantitative features, respectively, followed by a multivariable Cox regression analysis. We restricted the multivariable model to two features because of the small number of cases available. Using a 2D visualization of the data, we determined cut-off values to stratify patients into a low-risk group and a high-risk group for CLR. The resulting prognostic value was illustrated by means of Kaplan-Meier curves. We then characterized the low- and a high-risk groups for CLR using Mann-Whitney tests (for quantitative features) and Chi2 or Fisher exact tests (for clinical qualitative features). All of the statistical analyses were performed using Statistica software (StatSoft, Tulsa, OK, USA).

## Results

### Marker expression profiles in PCa versus PNT and in stroma versus epithelium

The present cohort of patients partially overlaps with the Gleason score (GS) 7 group in the previously described cohort [19]. Statistical analyses confirmed the previously described results

**Table 3. Summary of observed variations in marker expression.**

| Marker | N | PNT | P-value | N | PCa | P-value |
|---|---|---|---|---|---|---|
| | | Stroma < Epith | | | Stroma < Epith | |
| AR | 34 | 91% | < 0.001 | 47 | 100% | < 0.001 |
| ER | 23 | 0% | < 0.001 | 35 | 0% | < 0.001 |
| **Marker** | **N** | **Epithelium** | **P-value** | **N** | **Stroma** | **P-value** |
| | | PNT < PCa | | | PNT < Pca | |
| AR | 34 | 68% | 0.059 | 34 | 18% | < 0.001 |
| ER | 34 | 56% | 0.607 | 35 | 69% | 0.043 |
| PR | / | / | / | 35 | 31% | 0.043 |
| CD34 | / | / | / | 35 | 77% | 0.002 |
| αSMA | / | / | / | 34 | 74% | 0.010 |
| CD31 | / | / | / | 35 | 63% | 0.176 |
| CAV1 | / | / | / | 35 | 51% | 1.000 |

"Stroma < Epith" indicates the percentage of cases for which the LI was smaller in the stroma than in the epithelium areas from the same patient (applicable only for AR and ER). "PNT < PCA" indicates the percentage of cases for which the LI was smaller in the PNT than in the PCA samples from the same patient. The associated p-values resulted from the sign test.

concerning marker expression profiles in PCa versus PNT and in stroma versus epithelium, generally with more significant results because the current cohort is larger than the previous one. These results were summarized in Table 3, Fig 2 shows illustrative images for AR, αSMA and CD34, and Fig 3A illustrates the significant difference observed for αSMA, a stromal marker with potential prognostic impact (see next section). A new result obtained in the current enlarged GS7 cohort concerns the comparison between ISUP GG2 (GS 3 + 4) and ISUP GG3 (GS 4 +3). These two sub-groups, with quite similar profiles for the markers analyzed, showed differences for the stromal PR LI in PCa samples, which was higher in GG3 compared to GG2 (Mann-Whitney test: p = 0.0077, Fig 3B).

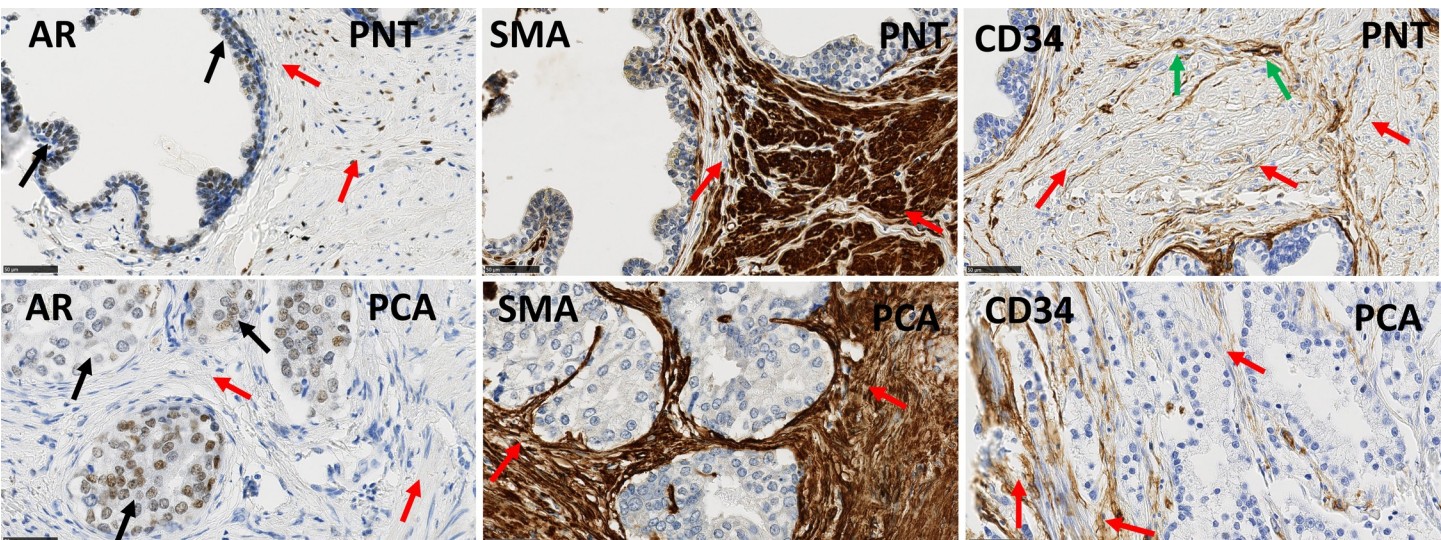

**Fig 2. Immunohistochemical stains for AR, αSMA and CD34 in PNT samples (top) and PCa samples (bottom) from GS7 patients.** Black arrows indicate epithelial cells and red arrows indicate stromal cells. AR is expressed on epithelial (black arrows) and stromal cells (red arrows). CD34 is also expressed in endothelial cells (green arrows). αSMA and CD34 show an increased stromal expression in a PCA sample compared to the paired PNT sample from the same patient. Scale bar equals 50μm.

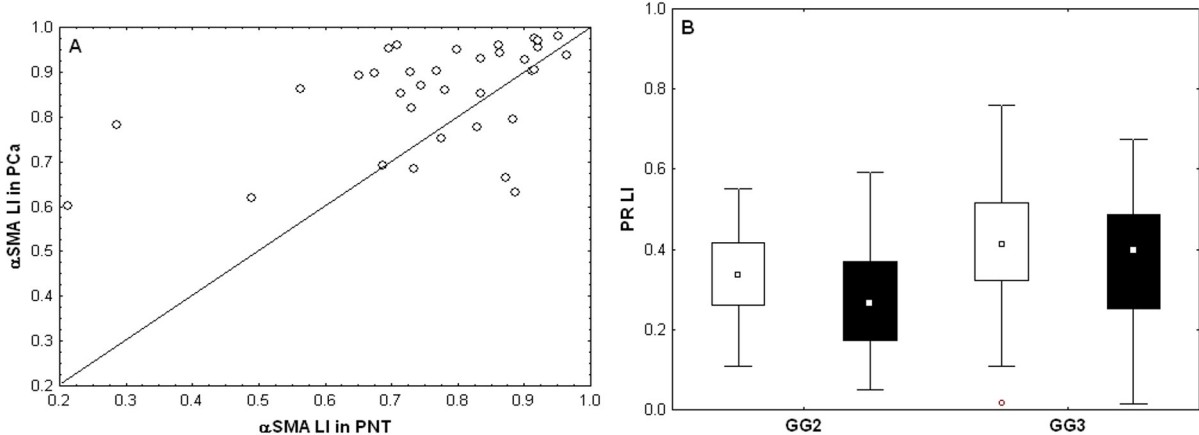

**Fig 3. Variations of quantitative expression levels (measured by their LI) of αSMA and PR biomarkers.** A) Paired αSMA expression levels in PNT (X-axis) and PCa (Y-axis) samples (each patient is represented by a dot; dots above the diagonal evidence increased expression levels in PCa samples). B) Stromal PR expression levels (LI) in PNT (white boxes) and PCa (black boxes) for patients in the GG2 and GG3 groups.

## CD31 and PR expression contribute to stratify PCa patients into two prognostic groups

It should be noted that because of the matched cohorts, no clinicopathological features had a significant prognostic impact (Table 1). This allowed us to focus our prognostic analysis on the biomarkers only.

Using univariable analyses of quantitative IHC values we identified several stromal markers that had a potential impact on clinical recurrence-free survival (CRFS, Table 4). The best p-values (i.e. < 0.10) were found for LI values evaluating CD31 expression in non-tumour (PNT) samples (HR = 9.833E+9, p = 0.007) and αSMA expression in PCa (HR = 25.223, p = 0.077) and for ratios of PR LI values evaluated in PCa and PNT stroma samples from the same patient (HR = 3.229, p = 0.031). For each of these three features a high value is associated with a higher risk of recurrence (HR greater than 1). It should be noted that the IHC features characterizing the PNT samples (and thus also the PCa/PNT ratios) concern a smaller number of patients due to reduced availability of normal tissue.

Table 5 details the multivariable model combining the two most significant features from Table 3 and shows that these can be considered as two independent prognostic features in terms of CLR (p-values < 0.05). We also tested other bivariable models based on the other pairs of features extracted from Table 2 and obtained less significant results. This multivariable analysis enabled us to identify a patient subgroup with highly significant different outcomes characterized by a lower risk (LR) of recurrence. This group is characterised by low expression levels of CD31 in non-tumour stroma (LI < 6%) and low ratios of tumour to non-tumour stromal PR expression levels (LI ratio < 80%), meaning that stromal PR expression is lower in

**Table 4. Univariable Cox regression analysis.**

| N | Risk factor | b | HR | 95% | CI | p-value |
|---|---|---|---|---|---|---|
| 35 | CD31 LI in PNT | 22.104 | 3.976E+9 | 200.162 | 7.898E+16 | 0.0099 |
| 48 | αSMA LI in PCA | 3.137 | 23.024 | 0.640 | 828.325 | 0.0862 |
| 35 | PR LI PCA/PNT | 1.111 | 3.037 | 1.042 | 8.852 | 0.0418 |

b: coefficient, HR: Hazard ratio, 95% CI: 95% of confidence interval, selection of features with p < 0.10.

**Table 5. Multivariable Cox regression model.**

| Model p-value | Risk factor | b | HR | 95% | CI | p-value |
|---|---|---|---|---|---|---|
| 0.0075 | CD31 LI in PNT | 26.365 | 2.818E+11 | 984.556 | 8.068E+19 | 0.0080 |
| (n = 35) | PR LI PCa/PNT | 1.174 | 3.234 | 1.133 | 9.228 | 0.0283 |

b: coefficient, HR: Hazard ratio, 95% CI: 95% of confidence interval

PCa compared to PNT samples from the same patient (Figs 4 and 5). The other patients constitute a group with a higher risk (HR) of recurrence (Fig 5).

We then verified the absence of bias in the above patients' stratification in terms of the clinical variables described in Table 1. No significant differences were evidenced for these variables between the LR and HR groups (see S1 Table). Finally, we characterized these prognostic groups in terms of the IHC markers which were not involved in the group identification (i.e. other than CD31 and PR). Only AR features exhibited significant variations as follows (Fig 6). The AR LI values evaluated in PCa stroma were higher in the HR group (p = 0.012), the ratios between the AR LI values evaluated in PCa and PNT stroma and in PCa and PNT epithelium from the same patient increased in HR (p = 0.022 and 0.006, respectively), whereas ratios between AR LI values evaluated in epithelial and stromal PNT samples from the same patient decreased in HR (p = 0.011).

## Discussion

Previously, we investigated the relation between different GS and the expression of several TME cell markers [19]. In the present study we investigated the potential of the same panel of TME cell markers to predict CLR in PCa. As detailed before [19], the panel of TME cell markers was composed based on their possible biomarker potential and/or clinical relevance, a high

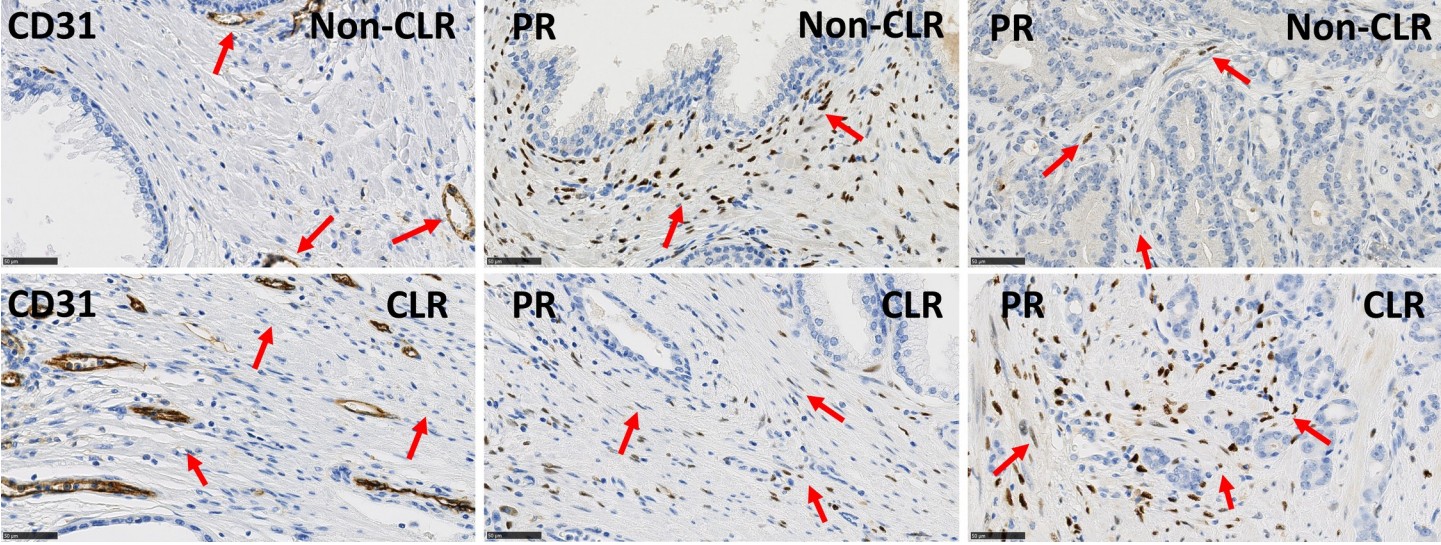

**Fig 4. Immunohistochemical stains for CD31 in PNT samples and for PR in PNT and PCa samples.** Upper panels illustrate a sample from the favourable PCa subgroup (non-CLR) with low risk to progression showing low expression levels of CD31 (vascular marker) in PNT stroma and low ratios of PR in PCa stroma/PR in PNT stroma. Lower panels illustrate a sample from the poor prognostic PCa subgroup (CLR) with high risk to progression showing high expression levels of CD31 (vascular marker) in PNT stroma and high ratios of PR levels in PCa stroma/PR levels in PNT stroma. Red arrows indicate vessels with expression of CD31 and stromal cells with expression of PR. Scale bar equals 50μm.

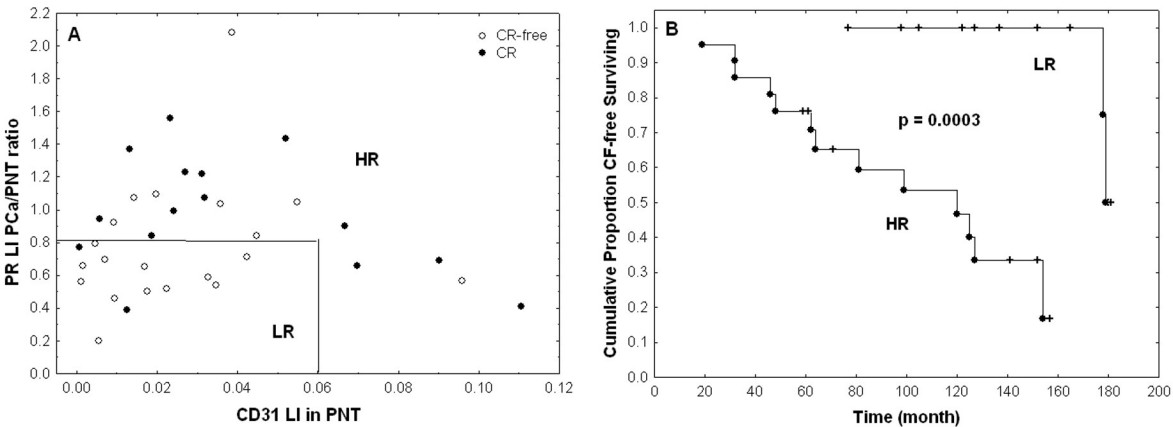

**Fig 5. Patient stratification on the basis of CD31 and PR expression levels.** A) A low risk (LR) group (n = 14) consists in samples combining low CD31 expression levels in PNT stroma (LI < 6%) and low ratios of PCa to PNT stromal PR expression levels (LI ratio < 80%). The other cases are grouped into a high risk (HR) group (n = 22). Open and black dots represent samples associated with absence and presence of CLR, respectively. B) Kaplan-Meier curves characterizing the LR and HR groups and showing highly significant difference in terms of CLR (the p-value is provided by the log-rank test).

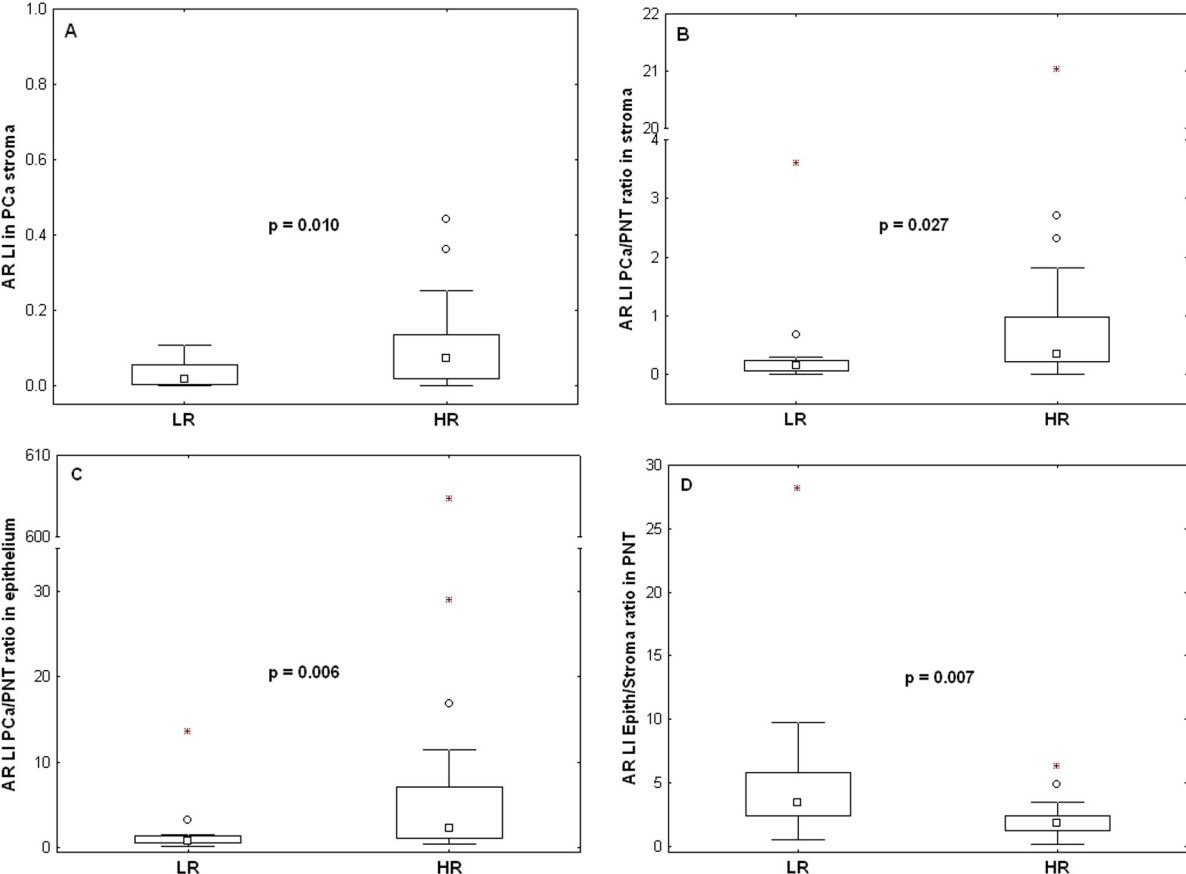

**Fig 6. Variations of AR expression levels (LI) between the low risk (LR) and the high risk (HR) groups determined in Fig 5.** A) Stromal AR expression levels in PCa. B) Ratio of PCa to PNT stromal AR expression levels. C) Ratio of PCa to PNT epithelial AR expression levels. D) Ratio of epithelial to stromal AR expression levels in PNT. The data distributions are described by means of their median (small square), interquartile range (box), non-outlier minimum and maximum values (bars) and the remaining outlier (dot) and extreme (asterisk) values. The p-values are provided by Mann-Whitney tests.

prevalence in PCa research, and the availability of reliable and specific antibodies. The central aim of this study was to look for prognostic subgroups in matched PCa cohorts depending on a specific TME cell profile. Such finding would support the importance of the TME in PCa progression and could potentially constitute a prognostic tool to aid in PCa patient risk stratification.

Univariable survival analyses revealed several markers with potential to predict CLR. High LI values of CD31 (vascular marker) in PNT stroma, high LI values of αSMA in PCa stroma and high PCa/PNT ratios of stromal PR LI values were found to be associated with increased risks of CLR. For all the other markers (AR, ER, CD34 and Cav1) no potential to predict CLR was found. Most of these stromal cell markers have been investigated separately in previous studies on their correlation with clinical outcome, but to our knowledge this is the first study looking at the prognostic potential of a multilayered panel of these stromal cell markers. A multivariable survival analysis distinguished a favorable PCa subgroup with LR to CLR progression. This group was identified by low LI values of CD31 (vascular marker) in PNT stroma and low PCa/PNT ratios of stromal PR LI values. The observation that alterations in the PNT stroma are also involved in the characterization of these favorable and unfavorable prognosis subgroups is intriguing, and supports the thesis of the so-called field effect cancerization.

IHC stains for vascular proteins are often used to study angiogenesis and evaluating microvessel density, an important parameter in cancer research, which is directly related to our quantitative CD31 LI feature. Many studies have looked at the relation between IHC expression of vascular proteins and CLR in PCa, and several of them have found a positive correlation [21–24] whereas others did not [25, 26]. While we were unable to find a relation between CD31 expression in PCa samples and CLR risk, we found that high CD31 LI values in PNT stroma are associated with an increased risk of CLR. It has been shown that the choice of vascular marker and the method of its evaluation–which vary in different studies–are critical for evaluating the angiogenic status [27]. Based on existing literature we have used CD31, proven to be a reliable vascular marker [26, 27], and image analysis for providing an objective evaluation of CD31 expression levels.

Functional conclusions cannot be drawn based on the present study design, but the finding that some of the studied TME markers have the potential to predict CLR is likely to reflect differences in stromal cell biology. Many studies have focussed on the role of the TME in PCa progression, and especially the role of SHR has drawn the attention of scientists [28]. Where we observed an association between high PCa/PNT ratios of stromal PR expression and an increased risk of CR, others did not [29–31]. The role of stromal AR in PCa progression is also still under debate. We were unable to find a significant relation between stromal AR and risk of recurrence. Some studies reported a correlation between higher AR in PCa stroma and disease progression [32, 33], which contrasts with other studies showing an association between lower AR in PCa stroma and disease progression and/or worse outcome [14, 15, 34, 35]. This underscores that the exact role of the reactive tumour stroma in modulating tumour progression is still under debate, although in general several studies suggest that the damage response biology of reactive stroma is likely to be tumour-promoting [36]. As discussed below, the seemingly contradictory results reported above may be due to multiple and different causes.

In general, literature often delivers opposing data on the relation between stromal cell markers and clinical outcome, and it is obvious that differences in methodologies and study design are at least partially responsible for this. First, we have applied a standardized digital image analysis. The advantages compared to the classic semi-quantitative assessment of IHC have been described comprehensively by our group and by several others [19, 37–42]. Second, it is crucial to look at the criteria used for disease progression and worse clinical outcome. We used time to CLR as parameter for disease progression (CLR was defined as established local

and/or distant disease recurrence with minimal follow-up time was 5 years), whilst others used cancer specific death/cancer specific survival or biochemical recurrence as primary endpoints [14, 15, 35]. Such heterogeneity in criteria for disease progression is common in literature and comparison of data should therefore always be related to the specific clinical endpoints. Third, we used matched patient cohorts (with and without CLR) in our study, whilst in other studies only single patient cohorts with disease progression were used [14, 15, 34, 35]. Fourth, most other studies used a wide GS range [14, 15, 34, 35], where we restricted our patient cohorts to GS7 (GG2-GG3) to minimize the bias of GS on clinical outcome. Indeed, we and others previously showed an association between stromal cell marker expression and a higher GS [15, 19, 34, 43], indicating a possible bias for GS in those studies without GS restriction.

The strengths of our study are the use of matched patient cohorts (with and without CLR), the use of a well-validated panel of stromal cell markers and the use of whole slide imaging and automated image analysis tools to characterize IHC. Weaknesses of our study are the relatively small sample sizes and the lack of prospective validation of our prognostic model in an independent patient cohort. The size of our patient cohorts was limited due to stringent matching criteria and given the large amount of marker combinations needed to be tested per sample. The two-marker prognostic model generated by the multivariable analysis in the present study will enable us to test this prospectively in a future large independent patient cohort. The exact role of the reactive tumour stroma in modulating PCa progression is still being discussed, but there is a growing amount of studies looking at the prognostic utilization of the stroma [35, 44–47]. Of interest, most commercially available prognostic genetic tests include several stromal cell markers in their gene panels [2]. However, only the Decipher-test includes a stromal cell marker that was used in the present study (Cav-1) [48], whilst Oncotype Dx [49] and Polaris (cell cycle progression signatures) [50] do not. These commercial genetic tests also have drawbacks: these are very expensive and rely on RNA-data, and may therefore miss relevant functional information at the protein level. IHC has the advantage of being a low-cost technique that targets proteins and captures both localization and heterogeneity of expression, but it has the disadvantage of covering only a limited amount of antigens, although multiplexing is gradually overcoming this problem. The challenge for future research will be the alignment and validation of the different prognostic stromal cell models in well-designed prospective studies. Multiplex IHC technology will be very useful if implemented at the clinical level.

In conclusion, this study shows a significant potential for several TME markers (CD31 in PNT stroma and PCa/PNT ratio of stromal PR expression) to predict CLR in matched PCa cohorts. Remarkably, the data also suggested a significant value for some TME markers in the stroma surrounding PNT to predict CLR. A multivariable survival analysis allowed us to identify an independent prognostic subgroup with significantly lower risk to CLR. If validated and confirmed in an independent prospective series, this subgroup might have clinical potential for PCa patient stratification and for guidance towards adjuvant, salvage or palliative treatment strategies. The present data further support a pivotal role for the TME in PCa progression.

## Supporting information

**S1 Table. Clinical and pathological characteristics of the LR and HR groups.**
(DOCX)

## Acknowledgments

The authors thank Justine Allard and Mélanie Derock (Diapath/CMMI) and Kathleen Van Den Eynde and Nathalie Volders (KU Leuven) for their excellent technical assistance.

## Author Contributions

**Conceptualization:** Thomas Gevaert, Thomas Vanden Broeck, Christine Decaestecker.

**Data curation:** Yves-Rémi Van Eycke.

**Formal analysis:** Yves-Rémi Van Eycke, Thomas Vanden Broeck, Sandrine Rorive, Christine Decaestecker.

**Funding acquisition:** Hein Van Poppel, Isabelle Salmon, Dirk De Ridder, Steven Joniau, Christine Decaestecker.

**Investigation:** Thomas Gevaert, Yves-Rémi Van Eycke.

**Methodology:** Thomas Gevaert, Yves-Rémi Van Eycke, Tim Muilwijk.

**Project administration:** Christine Decaestecker.

**Resources:** Hein Van Poppel, Dirk De Ridder.

**Software:** Yves-Rémi Van Eycke.

**Supervision:** Isabelle Salmon, Frank Claessens, Dirk De Ridder, Steven Joniau, Christine Decaestecker.

**Validation:** Isabelle Salmon, Sandrine Rorive, Frank Claessens, Steven Joniau.

**Visualization:** Yves-Rémi Van Eycke, Tim Muilwijk.

**Writing – original draft:** Thomas Gevaert, Tim Muilwijk.

**Writing – review & editing:** Thomas Gevaert, Christine Decaestecker.

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
