## [Decision Letter · Decision Letter 0]

12 Oct 2020

PONE-D-20-28259

The potential of tumour microenvironment markers to stratify the risk of recurrence in prostate cancer patients

PLOS ONE

Dear Dr. Gevaert,

Thank you for submitting your manuscript to PLOS ONE. After careful consideration, we feel that it has merit but does not fully meet PLOS ONE’s publication criteria as it currently stands. Therefore, we invite you to submit a revised version of the manuscript that addresses the points raised during the review process.

As you can realize both Reviewer ask for some further clarifications that would notably increase the strength of this paper. Furthermore, data from studies to be made freely available and this is not done in this case, even though it is said that it was.  Providing the data for this study would be as simple as a spreadsheet with each patient tumor score, stage, margin status, followup time to recurrence and stain scores  the analyzed patients

We look forward to receiving your revised manuscript.

Kind regards,

Antimo Migliaccio, M.D.

Academic Editor

PLOS ONE

Journal Requirements:

2. In the ethics statement in the manuscript and in the online submission form, please provide additional information about the patient samples used in your retrospective study, including: a) whether all data were fully anonymized before you accessed them; b) the date range (month and year) during which patients' medical records/samples were accessed and/or whether the IRB or ethics committee waived the requirement for informed consent; c) the date range (month and year) during which patients whose medical records/samples were selected for this study sought treatment; and d) the source of the medical records/samples analyzed in this work (e.g. hospital, institution or medical center name). If patients provided informed written consent to have data from their medical records used in research, please include this information.

3. Please note that PLOS does not permit references to “data not shown” or "not described" Authors should provide the relevant data within the manuscript, the Supporting Information files, or in a public repository. If the data are not a core part of the research study being presented, we ask that authors remove any references to these data.

4. At this time, we ask that you please provide scale bars on the microscopy images presented in Figures 1, 2 and 4, and refer to the scale bar in the corresponding Figure legend.

Reviewers' comments:

Reviewer's Responses to Questions

**Comments to the Author**

1. Is the manuscript technically sound, and do the data support the conclusions?

Reviewer #1: Yes

Reviewer #2: Partly

2. Has the statistical analysis been performed appropriately and rigorously? 

Reviewer #1: Yes

Reviewer #2: I Don't Know

3. Have the authors made all data underlying the findings in their manuscript fully available?

Reviewer #1: Yes

Reviewer #2: No

4. Is the manuscript presented in an intelligible fashion and written in standard English?

Reviewer #1: No

Reviewer #2: Yes

5. Review Comments to the Author

Reviewer #1: The manuscript by Dr. Thomas Gevaert and colleagues is interesting since it is very important to have personalised medicine for every patient. In this paper, the authors consider also the tumor microenvironment (TME) of the prostate cancer, analysing different markers in samples derived form Patients affected by Prostate Cancer (PC). This is an emerging and appealing area of interest. They investigated different TME-markers that possibly could predict CLR in PCa.

The manuscript is appealing and a good starting point since the knowledge of the tumor microenvironment is still limited to few studies, but some English style corrections are needed.

Furthermore, I have some considerations.

Materials and Methods:

1. line 135. Are the methods similar or the same? Please improve this sentence and add information.

Results:

2.In table 1 is not visible the Gleason's score, but the Gleason's grade, while in the paragraph "Marker expression profiles in PCa versus PNT and in stroma versus epithelium" the authors cite the Gleason's score. Please add information to the table or make everything more homogeneous and clear.

3. Is there also an important marker for TME, in particolar for Carcinoma associated fibroblasts. It is FAP-1. Have the authors evidences about its expression?

4. The authous should add a table resuming the first paragraph of the results and that could support the figure 1, also specifying if the proteins analyzed are present in the epithelial or the stromal compartments.

5. Discussion section:

The authors talk about the importance of TME in prostate cancer. They should take also in consideration the importance of steroid hormones receptors expression in this compartment, discussing about their absence/presence.

They could consider for example: 1) https://doi.org/10.3389/fendo.2014.00225; 2) Endocrine-Related Cancer

(2018) 25, R179–R196 and therein refs.

In general, the authors should improve the manuscript, the tables, the English grammar for enlarging the audience and rendering the manuscript more fluent.

Reviewer #2: This manuscript describes a biomarker panel of immunohistochemical stains that are prognostic of clinical recurrence after prostatectomy. The study was a "case-control" of 24 each of recurrent and nonrecurrent cases matched for Gleason score; it appears that attempts were made to match to other clinical features as well, but details for such matching are vague. Findings included that vascular marker CD31 in nontumor areas, alpha-smooth muscle actin staining in peritumor stromal tissue and cancer stroma / normal tissue stroma progesterone receptor staining ratio were all significant predictors of recurrence on univariable analysis, while a CD31 and PR stain ratio panel could break the study population into high and low risk groups for recurrence on multivariable analysis.

Strengths of study include a hard endpoint of image-verified tumor progression, an objective image analysis approach for stain scoring and an appropriate group of IHC markers directed towards TME proteins. Also it appears that the TMAs may have been meticulously designed however some details in support of this are lacking, and should be added to further support this idea. Lastly, the data suggest that the markers under study do have value, make biologic sense and deserve further study.

Weaknesses of the study include that the cohort is small. Also, Gleason scoring is known as the strongest prognosticator of biochemical recurrence after prostatectomy (after margin status), and in fact literature shows a large gap between GG2 and GG3 with GG3 essentially comparable to GG4, yet the authors have a cohort where the outcomes for GG2 and GG3 are similar. This is very disturbing and might put some readers off as to the value of these findings, although it could be argued that it further supports the morphologic and clinical homogeneity of the cohort under study. Another weakness is that while it is stated that the cohort is relatively homogeneous in clinical, stage, grade features, differing only in outcomes, data supporting this claim seems vaguely presented and should be more clearly presented.

Particular issues for authors to consider to improve the manuscript:

Introduction: The biomarkers discussed in the intro deal with predictive markers for treating metastatic disease and have nothing to do with the work presented. Biomarker panels (oncotype, decipher, prolaris, etc) for prognosticating recurrence are discussed in the same referenced paper in the intro and do deal with the same types of cases presented here and this should really be the focus of the intro. In fact, the impetus for doing a study like this would be to develop a significantly cheaper panel for doing the same job as these panels do, but at a significantly cheaper price. Also, this should be extended to the discussion since all of these panels also include stromal (TME) markers within the panels, so how this IHC panel does or does not include these /compare to what is in commercial panels would be very informative.

Methods: TMA core size should be indicated (.6mm, 1mm or other?). It is unclear how many cores of tumor per case and normal per case were included in the TMAs and it is mentioned that there were 6 cores taken and 3 TMAs made, but unclear if it is 3 copies of the same TMA design or that it took 3 sperate TMAs to get all of the cores included? 6 cores with three tumor / three normal? Other combo? Unclear, please clarify. While these details may seem mundane, there are some investigators that believe that at least 3 tumor cores are needed per case to account for tumor heterogeneity (for example see Rubin, Pienta, et all from around 2000 or 2002 for details), although a sentence of why such triplicates are not needed for this study is also acceptable. Also, as the makeup of the stroma might be different depending on where in the normal it is taken from (peripheral, central, transistional, periurethral, how close or far from the tumors), was there any effort to be consistent about where the normal was sampled from??? How close or far from the tumor was the normal taken? This should be included somewhere since there is discussion of the tumor field effect in discussion section.

Table 1, you really should have p values for each to be prognostic of clinical recurrence, because that would indicate whether clinical variables alone are prognostic and then whether the cohort needs to be reconsidered/redesigned.

In relation to this would be something about the details of the CLRs, how many were distant (bone or soft tissue mets) vs. how many were local failures (prostate bed or local LN mets), the latter of which might be more a function of surgical technique rather than actual tumor biology. Including such information could further help the reader put these results into perspective.

6. PLOS authors have the option to publish the peer review history of their article (what does this mean?). If published, this will include your full peer review and any attached files.

Reviewer #1: No

Reviewer #2: No

---

## [Author Response · Author response to Decision Letter 0]

6 Nov 2020

We thank the reviewers and editor for the interesting and adequate comments and we have carefully answered and addressed all of the reviewers’ and editors’ comments. 

# Editor:

Answer: Done.

2. In the ethics statement in the manuscript and in the online submission form, please provide additional information about the patient samples used in your retrospective study, including: a) whether all data were fully anonymized before you accessed them; b) the date range (month and year) during which patients' medical records/samples were accessed and/or whether the IRB or ethics committee waived the requirement for informed consent; c) the date range (month and year) during which patients whose medical records/samples were selected for this study sought treatment; and d) the source of the medical records/samples analyzed in this work (e.g. hospital, institution or medical center name). If patients provided informed written consent to have data from their medical records used in research, please include this information.

Answer: Done.

3. Please note that PLOS does not permit references to “data not shown” or "not described" Authors should provide the relevant data within the manuscript, the Supporting Information files, or in a public repository. If the data are not a core part of the research study being presented, we ask that authors remove any references to these data.

Answer: We have included a supplementary table S1.

4. At this time, we ask that you please provide scale bars on the microscopy images presented in Figures 1, 2 and 4, and refer to the scale bar in the corresponding Figure legend

Answer: Scale bars (50µm) for figures 1, 2 and 4 are added and referred to in the corresponding figure legends.

# Reviewer 1:

1. English language: 

Answer: The manuscript had been revised for English grammar and adapted accordingly.

2. Materials and Methods: line 135. Are the methods similar or the same? Please improve this sentence and add information:

Answer: This has been addressed in the revised manuscript.

3. Results: In table 1 is not visible the Gleason's score, but the Gleason's grade, while in the paragraph "Marker expression profiles in PCa versus PNT and in stroma versus epithelium" the authors cite the Gleason's score. Please add information to the table or make everything more homogeneous and clear.

Answer: This should be: ‘ISUP Grade Group’ and has been changed accordingly. ‘ISUP Grade Group ‘ and Gleason Score are the two accepted terminologies to categorize prostate cancer grades.

4. Is there also an important marker for TME, in particolar for Carcinoma associated fibroblasts. It is FAP-1. Have the authors evidences about its expression?

Answer: This is an interesting remark form the reviewer. We did not check this one (also since this study has been designed with a similar marker panel as previously used (Gevaert el al. Scientific Reports, 2018). We will consider including FAP-1 in future research projects.

5. The authors should add a table resuming the first paragraph of the results and that could support the figure 1, also specifying if the proteins analyzed are present in the epithelial or the stromal compartments.

Answer: A table has been added accordingly in the revised manuscript (Table 3).

6. The authors talk about the importance of TME in prostate cancer. They should take also in consideration the importance of steroid hormones receptors expression in this compartment, discussing about their absence/presence. They could consider for example: 1) https://doi.org/10.3389/fendo.2014.00225;) Endocrine-Related Cancer

(2018) 25, R179–R196 and therein refs.

Answer: The possible role of stromal steroid hormone receptors in PCa progression is addressed in the discussion section and we also have referenced the proposed paper (and others).

7. In general, the authors should improve the manuscript, the tables, the English grammar for enlarging the audience and rendering the manuscript more fluent.

Answer: The manuscript has been revised and improved, tables have been improved + new tables are included , and the paper has been revised for English grammar.

Reviewer #2: 

1. Introduction: 

The biomarkers discussed in the intro deal with predictive markers for treating metastatic disease and have nothing to do with the work presented. Biomarker panels (oncotype, decipher, prolaris, etc) for prognosticating recurrence are discussed in the same referenced paper in the intro and do deal with the same types of cases presented here and this should really be the focus of the intro. In fact, the impetus for doing a study like this would be to develop a significantly cheaper panel for doing the same job as these panels do, but at a significantly cheaper price. Also, this should be extended to the discussion since all of these panels also include stromal (TME) markers within the panels, so how this IHC panel does or does not include these /compare to what is in commercial panels would be very informative.

Answer: 

The introduction and discussion sections have been revised and amended accordingly. The commercially available prognostic biomarker panels are discussed related to our panel and related to the pro’s and con’s of IHC.

2. Methods: 

- TMA core size should be indicated (.6mm, 1mm or other?): 

Answer: 6mm, changed accordingly.

- It is unclear how many cores of tumor per case and normal per case were included in the TMAs and it is mentioned that there were 6 cores taken and 3 TMAs made, but unclear if it is 3 copies of the same TMA design or that it took 3 sperate TMAs to get all of the cores included? 6 cores with three tumor / three normal? Other combo? Unclear, please clarify. While these details may seem mundane, there are some investigators that believe that at least 3 tumor cores are needed per case to account for tumor heterogeneity (for example see Rubin, Pienta, et all from around 2000 or 2002 for details), although a sentence of why such triplicates are not needed for this study is also acceptable. 

Answer: Three TMAs were required to include all cores from the 48 patients. Per patient, 6 tumor cores and 6 normal tissue cores were included. This has been clarified in the revised manuscript. 

- Also, as the makeup of the stroma might be different depending on where in the normal it is taken from (peripheral, central, transistional, periurethral, how close or far from the tumors), was there any effort to be consistent about where the normal was sampled from??? How close or far from the tumor was the normal taken? This should be included somewhere since there is discussion of the tumor field effect in discussion section.

Answer: Since almost all tumours (ISUP GG2-3) were peripheral zone tumours, control tissue was taken from the contralateral non-tumour peripheral zone. This has been added to the methods section.

3. Table 1, you really should have p values for each to be prognostic of clinical recurrence, because that would indicate whether clinical variables alone are prognostic and then whether the cohort needs to be reconsidered/redesigned.

Answer: The p-values for evaluating the prognostic impact were added in Table 1. Because of the matched cohorts, p-values > 0.05 were expected.

4. In relation to this would be something about the details of the CLRs, how many were distant (bone or soft tissue mets) vs. how many were local failures (prostate bed or local LN mets), the latter of which might be more a function of surgical technique rather than actual tumor biology. Including such information could further help the reader put these results into perspective.

Answer: These details on CLRs are now included in the methods section. Of note, patient groups were matched for section margins and lymph node status, parameters which are likely to contribute to CLR (see also previous comment).

---

## [Decision Letter · Decision Letter 1]

15 Dec 2020

The potential of tumour microenvironment markers to stratify the risk of recurrence in prostate cancer patients

PONE-D-20-28259R1

Dear Dr. Gevaert,

We’re pleased to inform you that your manuscript has been judged scientifically suitable for publication and will be formally accepted for publication once it meets all outstanding technical requirements.

Kind regards,

Antimo Migliaccio, M.D.

Academic Editor

PLOS ONE

Additional Editor Comments (optional):

Reviewers' comments:

Reviewer's Responses to Questions

**Comments to the Author**

1. If the authors have adequately addressed your comments raised in a previous round of review and you feel that this manuscript is now acceptable for publication, you may indicate that here to bypass the “Comments to the Author” section, enter your conflict of interest statement in the “Confidential to Editor” section, and submit your "Accept" recommendation.

Reviewer #1: All comments have been addressed

2. Is the manuscript technically sound, and do the data support the conclusions?

Reviewer #1: Yes

3. Has the statistical analysis been performed appropriately and rigorously? 

Reviewer #1: Yes

4. Have the authors made all data underlying the findings in their manuscript fully available?

Reviewer #1: Yes

5. Is the manuscript presented in an intelligible fashion and written in standard English?

Reviewer #1: Yes

6. Review Comments to the Author

Reviewer #1: The authors have addressed most of my questions and have written some concepts in a more attractive and understendable way. I think that addition of data in TME field is very important and in this form the manuscript is suitable for the publication.

7. PLOS authors have the option to publish the peer review history of their article (what does this mean?). If published, this will include your full peer review and any attached files.

Reviewer #1: No

---

## [Editor Report · Acceptance letter]

16 Dec 2020

PONE-D-20-28259R1 

The potential of tumour microenvironment markers to stratify the risk of recurrence in prostate cancer patients 

Dear Dr. Gevaert:

I'm pleased to inform you that your manuscript has been deemed suitable for publication in PLOS ONE. Congratulations! Your manuscript is now with our production department. 

Kind regards, 

on behalf of

Dr. Antimo Migliaccio 

Academic Editor

PLOS ONE